# Verification and Use of the US-FDA BAM 19b Method for Detection of *Cyclospora cayetanensis* in a Survey of Fresh Produce by CFIA Laboratory

**DOI:** 10.3390/microorganisms10030559

**Published:** 2022-03-04

**Authors:** Laura Lalonde, Jenna Oakley, Patrick Fries

**Affiliations:** Centre for Food-Borne and Animal Parasitology, Saskatoon Laboratory, Canadian Food Inspection Agency, 116 Veterinary Road, Saskatoon, SK S7N 2R3, Canada; jenna.oakley@inspection.gc.ca (J.O.); patrick.fries@inspection.gc.ca (P.F.)

**Keywords:** *Cyclospora*, food safety, leafy greens, berries, herbs, qPCR

## Abstract

To facilitate the harmonized surveillance and investigation of cyclosporiasis outbreaks in the US and Canada, we adapted and verified the US-FDA’s BAM 19b method and employed it in a national produce survey. Performance was verified by spiking 200, 10, 5 or 0 *C. cayetanensis* oocysts onto berries (50 ± 5 g, *n* = 85) and 200, 10 or 0 oocysts onto green onions (25 ± 3 g, *n* = 24) and leafy greens (25 ± 1 g, *n* = 120) and testing these samples by the BAM method on Bio-Rad CFX96. Method robustness was assessed by aging (0 or 7 days) and freezing the produce and washes prior to testing, then implementing the method for the surveillance testing of 1759 imported leafy green, herb and berry samples. Diagnostic sensitivity was 100/44% and 93/30% for berries and leafy greens spiked with 200/10 oocysts, respectively. The diagnostic and analytical specificity were 100% for all matrices and related parasites tested. The proportion positive was unaffected (*p* = 0.22) by age or condition of produce (7d, fresh, frozen) or wash concentrate (3d, fresh, frozen); however, the Cq values were higher (*p* = 0.009) for raspberries aged 7d (37.46 ± 0.29) compared to fresh (35.36 ± 0.29). *C. cayetanensis* was detected in berries (two), herbs (two) and leafy greens (one), representing 0.28% of the tested survey samples. These results independently verified the reported performance characteristics and robustness of the BAM method for the detection of *C. cayetanensis* in a variety of matrices, including under adverse sample conditions, using a unique detection platform and demonstrating its routine diagnostic use in our Canadian Food Inspection Agency (CFIA) laboratory.

## 1. Introduction

Multi-jurisdictional outbreaks of locally acquired *Cyclospora* cases have been investigated annually in Canada since 2013. These outbreaks occur seasonally, typically between May and August of each year. The number of locally acquired cases of cyclosporiasis reported in these outbreak investigation ranges has been steadily increasing, now totalling 1352 Canadian cases. Since 2015, 8018 cases of food-borne cyclosporiasis have been reported in the US, and the food vehicle or source of the infection has not been identified in any of these recent outbreaks [1,2]. To facilitate and improve disease outbreak investigations and the routine surveillance of produce for food-borne parasites, reliable, validated detection tools are critical. Timely testing is required to ensure that appropriate regulatory action can be taken while produce is still on the shelves or in the fridge of consumers. Too often, once testing is complete, the implicated produce is no longer available or deteriorated and not suitable for further testing and trace-back investigation. The detection of disease outbreaks and chronic health issues associated with protozoan parasite infection is difficult; since the onset of symptoms can vary from days to years (e.g., toxoplasmosis), many cases go unreported or underrecognized, and it is difficult to retroactively link cases with positive food samples. 

At the Centre for Food-borne and Animal Parasitology (CFAP), we performed routine surveillance of imported and domestic fresh produce for *C. cayetanensis* and other protozoan parasites like *Cryptosporidium*, *Toxoplasma* and *Giardia* using validated molecular methods for several years [3,4]. The qPCR assay with melt curve analysis (MCA) that we designed and employed can detect *Cyclospora*, *Cryptosporidium* and *Toxoplasma* [5,6]. The broad specificity of the primer cocktail used for the qPCR MCA is useful for detecting multiple species of coccidia but also sometimes results in preferential nonspecific amplification of fungal and other background DNA, which reduces the assay sensitivity and may complicate the interpretation of results. Therefore, a species-specific assay for the detection of *C. cayetanensis* would be more suitable, especially as a confirmatory test and for use in disease outbreak investigations. 

The US Food and Drug Administration (US-FDA) released the Microbiological Methods & Bacteriological Analytical Manual (BAM) Chapter 19b—‘Molecular Detection of *Cyclospora cayetanensis* in Fresh Produce Using Real-Time PCR’ in 2017 [7]. This method is reported to be *Cyclospora*-specific, has undergone interlaboratory validation and has a reported detection limit of five oocysts in cilantro and raspberries [8,9]. There is currently no international standard for the detection of *C. cayetanensis* in fresh produce; therefore, the International Organization for Standardization (ISO) is in the process of developing such a standard that could be based on several aspects of the BAM Chapter 19b protocol (hereinafter referred to as the BAM method). Considering the sensitivity limitations of our currently employed *Cyclospora* detection assay, and to facilitate a harmonized and effective approach to routine surveillance and the investigation of cyclosporiasis outbreaks in the US and Canada, the objective of this project was to adapt, verify and implement the BAM method for routine surveillance at our CFIA Saskatoon Laboratory.

## 2. Materials and Methods

### 2.1. Parasites

*C. cayetanensis* oocysts stored in 2.5% potassium dichromate were kindly provided by Dr. Alexandre da Silva, US-FDA in May 2018 for use in this study. To prepare dilutions for spiking, the oocysts were washed by centrifugation to remove potassium dichromate, enumerated by a haemocytometer and diluted to the required concentration (either 200, 10 or 5 oocysts/25 µL, as per the US-FDA validation study [9]) in 0.1% Alconox (Sigma-Aldrich, Oakville, ON, Canada) solution supplemented with 1× antibiotic–antimycotic solution (Gibco™, Waltham, MA, USA). The quantity of oocysts in the prepared spiking stocks was then verified by counting two 25-µL aliquots on a welled microscopy slide. *Cryptosporidium parvum* oocysts (calf-passaged, Iowa isolate) were purchased from Waterborne, Inc. (New Orleans, LA, USA) for use as the DNA extraction positive control. *C. parvum* oocysts were enumerated by the haemocytometer and stored at 4 °C in 1× antibiotic–antimycotic solution (Gibco).

### 2.2. Produce Samples

We verified the method for use in raspberries; strawberries; blueberries; cilantro; romaine lettuce; spring mix; kale salad mix (containing kale, cabbage, Brussel sprouts and chicory); coleslaw (containing red and green cabbage and shredded carrots) and whole green onions. These matrices were selected, since they have been commonly tested by the CFIA-CFAP in routine targeted surveillance for *C. cayetanensis*. The freshest available of these commodities were selected and purchased from local grocery stores for the spiking experiments. Produce was purchased a maximum of 24 h prior to use unless otherwise specified. Leafy greens (25 ± 1 g), green onions (25 ± 3 g) and berries (50 ± 5 g) were weighed directly into BagPage^®^+ filter bags (400-mL size with a microperforated filter; Cole-Parmer, Montreal, QC, Canada) and spiked with 5, 10 or 200 *C. cayetanensis* oocysts. Ten samples were tested at each spike level, and four un-spiked samples were tested for each produce type. Oocysts were applied directly onto the samples in 3 to 4 droplets using an air displacement variable volume pipette fitted with a plastic tip and allowed to dry for 2 to 3 h at room temperature. The bags were then sealed, and the samples were stored overnight (~18 h) at 4 °C prior to processing, as described below.

### 2.3. Method Modifications

All produce samples were processed according to the US-FDA’s BAM Chapter 19b method, with the following adaptations and modifications. Our centrifuges are equipped with buckets and inserts to spin 250-mL conical bottom polypropylene bottles (Corning, Corning, NY, USA), so we verified the procedure using these tubes in place of 2 × 50-mL tubes, as specified in the BAM method. This eliminates one centrifugation and transfer step. For verification experiments, an extraction positive control consisting of 200 *C. cayetanensis* oocysts in 50 µL of negative produce wash carrier and an extraction negative control consisting of extraction reagents only were included in each batch. We have observed in previous studies that the FastDNA™ Spin Kit for soil (MP Biomedicals, Irving, CA, USA) does not perform well for parasites suspended only in PBS [10]; a carrier (produce wash concentrate or 8% skim milk) is required for efficient extraction. For implementation in routine diagnostic testing, 2000 *C. parvum* oocysts in 8% skim milk were used as the extraction positive control and extraction reagents only as the negative control. Amplification of the extraction positive and negative controls was performed using *C. parvum*-specific JVAGF/R primers and the JVAGP2 probe [11] on the same qPCR plate alongside the test samples using the same master mix reagents. 

We verified our qPCR instrument (CFX96, Bio-Rad, Hercules, CA, USA) and Bio-Rad CFX Manager Software (version 3.1) for use with the BAM method, which was originally validated on the Applied Biosystems 7500 Fast Real-Time PCR System. The instrument software’s auto-calculate feature was used to determine the baseline threshold, and the baseline subtracted curve fit setting was enabled. A five-point standard curve consisting of ten-fold serial dilutions (4 × 10^5^ copies to 4 × 10^1^ copies) of the HMgBlock135m gene block (artificial *Cyclospora* positive control [7]) was included on each plate in duplicate.

### 2.4. qPCR Master Mix and Exclusivity Evaluation

Others have reported cross-reactivity from the primers for this assay, with the 18S rRNA gene in closely related coccidia species [12], although this occurred using a non-validated qPCR platform, reagents and probe labelling different from what is specified in the BAM method, which were selected based on the results from validation studies. Therefore, we evaluated the performance and specificity of the BAM qPCR assay on our Bio-Rad platform using the recommended mix: Quantifast (Mix 1, Qiagen, Toronto, ON, Canada), as well as Sso Advanced Universal Probes Super Mix (Mix 2, Bio-Rad), iTaq Universal Probes Super Mix (Mix 3, Bio-Rad), TaqMan™ Fast Advanced Master Mix (Mix 4, Applied Biosystems, Waltham, MA, USA) and TaqMan™ Environmental Master Mix (Mix 5, Applied Biosystems). We tested gDNA from *Toxoplasma gondii*, *Cystoisospora belli*, *Hammondia hammondi*, *Eimeria acervulina*, *Eimeria maxima*, *Eimeria necatrix*, *Eimeria tenella*, *Eimeria bovis*, *Eimeria ahsata*, *Eimeria papillata*, *Cryptosporidium parvum*, *Cryptosporidium hominis*, *Cryptosporidium meleagridis*, *Giardia muris*, *Giardia duodenalis* and *Isospora suis* with each mix under the same amplification conditions and compared the results. The DNA for each species in the exclusivity panel was extracted from 20,000–25,000 oocysts/cysts (500 cysts or oocysts/µL DNA) and was verified to be suitable for amplification by the qPCR-MCA assay [3] for coccidia or the LAMP assay [13] for *Giardia.*


### 2.5. Robustness and Repeatability 

The method’s robustness was evaluated in a 3 × 3 factorial experiment by testing spring mix and raspberry samples that had been subjected to various storage conditions expected during routine diagnostic testing. Spring mix and raspberries were selected as representative matrices for leafy greens and berries, each tending to be more fragile than other produce types. Produce samples were divided into three groups (*n* = 15/group) and stored at 4 °C overnight, frozen for 1–3 days or aged 7 days at 4 °C prior to spiking with 200 *C. cayetanensis* oocysts and processing as described above. For each group, the final produce wash either proceeded to DNA extraction immediately (*n* = 5), was stored at 4 °C (*n* = 5) or frozen (*n* = 5) for 2 days and then extracted. The impact of these changes on the Cq values and proportion positive was then calculated.

To determine the suitability of the BAM method for testing frozen berries contaminated with *C. cayetanensis*, we spiked samples of raspberries with 200 (*n* = 16), 10 (*n* = 16) or 0 (*n* = 4) oocysts. Half of these samples were processed fresh the day following spiking, and half were frozen at −20° C for 7 days, then thawed at room temperature for 30 min and processed. Proportion-positive and Cq values were compared for each group. 

To determine the degree of between-run repeatability of the results, we assessed the variability in the Cq values generated for the *Cyclospora* artificial positive control (10^5^ standard as above) on five different plates (runs) conducted by five different analysts on different days. The average Cq value and the coefficient of variation were calculated for all the replicates across all the runs. To determine the degree of within-run repeatability, we assessed the variability in the Cq values for replicates of the same sample on the same five plates as above.

### 2.6. Proficiency Testing

Our laboratory is accredited by the Standards Council of Canada (SCC) for testing in accordance with the International Organization for Standardization (ISO) 17025: 2017 standard and for research according to the SCC Requirements and Guidance for Accreditation of Laboratories Engaged in Test Method Development and Non-Routine Testing. As such, all method development and verification work was carried out in accordance with these requirements. To facilitate the implementation of the BAM method for routine surveillance testing of leafy greens, berries and herbs, we trained five analysts to perform the test and administered proficiency panels to each analyst. The proficiency panels consisted of six different produce samples, five of which were spiked with 200 *C. cayetanensis* oocysts and one randomly selected un-spiked sample. The panels were prepared as above for the method verification experiments and analyzed by five different analysts on different days. Each panel also included the *Cyclospora* positive control standard curve, a no-template control (NTC), a DNA extraction positive control (2000 *Cryptosporidium* oocysts) and a DNA extraction negative control (extraction reagents only), in addition to the six panel samples. For the proficiency panels and subsequent survey testing, all test samples were run in triplicate, and the standards and controls were run in duplicate. The analysts must correctly identify all panel samples as positive or negative, and all controls must perform as expected to be considered successfully qualified to perform the test in a diagnostic or research setting.

### 2.7. Implementation

The BAM method (adapted as above for use in our laboratory) was employed for use in the CFIA’s food microbiological surveillance program, which consists of the National Microbiological Monitoring Program (NMMP) and the targeted surveys program. Under both programs, a variety of domestic and imported products are randomly selected and tested for a variety of microbial organisms, including bacterial pathogens, viruses and parasites, including *Cyclospora*, in fresh produce. Samples of fresh whole and cut leafy vegetables and vegetable salads (653 samples, plans F263R, SB3221 and SB3222); leafy herbs (489 samples, plan SB3081) and berries (597 samples, plan SB 341) were collected at retail outlets in Alberta, British Colombia, Manitoba, New Brunswick, Nova Scotia, Ontario, Quebec, and Saskatchewan between July 2020 and 31 December 2021 and shipped to the CFIA Saskatoon laboratory. An additional 20 berry samples (each submission consisting of 5 units from the same lot) were collected at licensed produce facilities by CFIA inspectors for the NMMP sample plan (F227). Sample sizes, handling and quality control measures were as outlined in Lalonde and Gajadhar [3]. Samples identified as positive by qPCR were repeated without the internal amplification control (IAC) for both confirmation and to generate amplicons for sequencing (as below) and were reported as positive if at least 1/3 of the replicates showed a Cq value < 45 on both the first and subsequent qPCR.

### 2.8. Sequencing

During verification, selected qPCR amplicons from samples of cilantro and raspberries spiked with 200 *C. cayetanensis* oocysts were sequenced to confirm the identity of the target product. The amplicons for sequencing were generated by performing the qPCR on gDNA from these samples without the IAC to improve the probability of a successful sequencing reaction. The qPCR products were purified using the QIAquick PCR purification Kit (Qiagen) according to the manufacturer’s protocols and sequenced in both directions. Sequences of the forward and reverse reads were trimmed and assembled into contigs using Clone Manager 9 Professional (Sci-Ed Software) and compared to the available sequences in GenBank using NCBI-BLAST.

### 2.9. Statistics

For experiments of a factorial design, a two-way ANOVA was performed using GraphPad Prism (version 4.03) with an alpha level of 0.05, and significant difference was considered if the *p*-value < 0.05. For comparing means, an unpaired *t*-test was performed using GraphPad Prism, and significant difference was considered if the *p*-value < 0.05.

## 3. Results

### 3.1. qPCR Master Mix and Exclusivity Evaluation

We evaluated the performance and exclusivity of five different qPCR master mixes with the BAM method on our Bio-Rad CFX96 instrument (Table 1). Three of the five mixes evaluated amplified only *C. cayetanensis* gDNA and did not amplify DNA from any of the nontarget species. Two of the master mixes evaluated also amplified the DNA from *E. acervulina*, *E. maxima*, *E. necatrix*, *E. bovis*, *E. tenella*, *E. ahsata* and *I. suis* and were therefore not selected for further evaluation. The Quantifast reagent was selected for use in all subsequent verification experiments, as it is the master mix recommended in BAM Chapter 19b. 

We submitted qPCR amplicons generated from spiked produce samples for sequencing to verify the amplification of *C. cayetanensis* with the BAM primers/probe. However, due to the product’s short length, the sequencing of many of the amplicons failed outright, and none produced a quality assembled (forward and reverse) sequence longer than 50–60 bp. However, the BLAST analysis of the successful forward and reverse reads produced matches to *C. cayetanensis* with 98–100% percentage identity (data not shown). 

### 3.2. Verification in Berries, Leafy Greens and Herbs

We verified the suitability of the BAM method for use on four types of leafy greens (romaine, spring mix, coleslaw and kale salad mix); one herb (cilantro); three berries (strawberry, raspberry and blueberry) and one vegetable (green onions). The results of these spiking experiments are presented in Table 2. The average diagnostic sensitivity (DSe percentage of the spiked samples that were positive by qPCR) for the berries was 100% at the 200 oocyst spike level and 44% for samples spiked with 10 oocysts. The DSe was highest for strawberries, which were positive in 100, 82.8 and 27% of the samples spiked with 200, 10 and 5 oocysts, respectively. For cilantro, the DSe was 100% at the 200 spike level and 50% for samples spiked with 10 oocysts. For leafy greens, the average DSe was 92.5% for samples spiked with 200 oocysts and 25% for the 10 oocyst spike level. For green onions, the DSe was 90% for the 200 spike level and 30% for the 10 spike level. The diagnostic specificity (DSp percentage of the un-spiked samples that were negative by qPCR) was 100% for all matrices. The method detection limit was considered to be reached when between 25 and 75% of the spiked samples were positive, and the number of oocysts spiked was based on the US-FDA validation study [9]. Based on these verification experiments, we determined the detection limit of the BAM method to be 10 *C. cayetanensis* oocysts for green onions, blueberries, raspberries, kale salad mix, coleslaw and cilantro. The detection limit for spring mix, romaine and blueberries was lower (between 10 and 200 oocysts) and five oocysts for strawberries.

### 3.3. Robustness

The results for the robustness experiments on the spring mix and raspberries are presented in Table 3. For the spring mix, the Cq values were higher (not statistically significant) and detection rates lower when the produce was aged 7d. For raspberries, aging the samples increased the Cq value (*p* = 0.009); however, the detection rate remained at 100%, and freezing the berries did not impact either measure (*p* > 0.05). There was no impact of storing or freezing the produce wash prior to DNA extraction for either matrix.

### 3.4. Freezing

We examined the impact of freezing raspberries on the recovery of *C. cayetanensis* oocysts in situ. Raspberries were spiked with high (200) or low (10) numbers of oocysts and then either frozen at −20 °C for 7 days or processed fresh (Table 4). Freezing the spiked berries had no impact on the Cq values for the high or low spike groups. The proportion positive for frozen berries was lower than the fresh ones for the low spike group, although still comparable to the previously determined detection limit (Table 2).

### 3.5. Repeatability and Reproducibility

The within- and between-run repeatability experiment results are presented in Table 5. The average Cq value was 20.95 ± 0.30, and the coefficient of variation was 1.45% for all the replicates across all the runs. To determine the degree of within-run repeatability, we assessed the variability in the Cq values for replicates of the same sample on the same five plates as above. The coefficient of variation ranged from 0.91 to 2.72% between replicates.

The proficiency panel results are presented in Table 6. All positive and negative samples were correctly identified, and the Cq values for the positive samples were consistent (36.14 ± 1.33) with a CV of 3.67%, regardless of the produce matrix type.

### 3.6. Surveillance of Fresh Imported Produce

A summary of the number and types of fresh imported produce tested for *Cyclospora* in the NMMP and targeted survey program is presented in Table 7. *C. cayetanensis* was detected by the BAM method in samples of spinach (US), mint (Colombia), cilantro (US), blueberries (US) and raspberries (Mexico) (Table 8). The majority of samples positive for *Cyclospora* were detected in the fall/winter seasons (4/5); one was detected in early summer. One sample (mint) was classified as organic according to the package labelling, and the others were conventionally grown. The percentage positive for each produce type was highest for the herbs (2/489, 0.41%), followed by berries (2/617, 0.32%) and leafy greens (1/352, 0.28%).

## 4. Discussion

We have verified the performance of the BAM Chapter 19b method in our laboratory and implemented it for the testing of fresh leafy greens, herbs and berries for *C. cayetanensis*. The DSe reported in this study is similar to that obtained by the FDA for leafy greens and berries at the 200 spike level [9]. However, our reported DSe is lower than the FDA’s for the 10 and 5 spike levels, likely due to differences in the ages of the oocysts and the in procedures for preparation and enumeration of the spiking stocks. The FDA procedure for preparing the spike stock did not involve verification of the oocyst numbers after serial dilution to the required spiking concentration. We verified the number of oocysts in the actual spike aliquot by microscopy for each stock prepared. Additional factors that could have contributed to the lower sensitivity included the use of a different amplification platform and of an oocyst stock that was only 4% sporulated. The *C. cayetanensis* genome encodes multiple copies of the target SSU rDNA, and as the oocysts sporulate, the number of target gene copies increases, thus further enabling detection by qPCR.

The method has been verified in our laboratory for the produce types listed in Table 2. Due to their similarity in physical characteristics to those of the matrices listed, other soft-stemmed herbs such as parsley, basil, mint and dill (normally consumed raw) should be suitable for testing, as should other types of lettuces (e.g., green/red leaf and Swiss chard), chives and blackberries. Other new produce types with unique physical characteristics to the above will require verification via matrix extension studies. The BAM method has also been demonstrated as suitable for the detection of *Cyclospora* in prepared dishes such as salsa, guacamole and coleslaw, with dressing with some modifications [14,15]. This apparent versatility and adaptability of the method to a broad range of matrices will be useful in both surveillance and disease outbreak investigations.

We have demonstrated 100% DSp in our verification experiments, which is consistent with the interlaboratory validation study results reported by the FDA [9]. Two of the qPCR master mixes that we evaluated showed cross-reactivity with several *Eimeria* species and *I. suis*, which, although not pathogenic to humans, could be found in the environment or on produce. These qPCR mixes likely do not provide the level of stringency required for specific binding of the 18S rRNA target primers and probe. Others have reported cross-reactivity of the BAM method primers and/or probe with non-target species while using a different procedure that deviated from the protocol described in the BAM [12]. It is important to highlight that the protocol described in the BAM underwent a rigorous multi-laboratory validation study [9]. Therefore, to avoid any bias, it is important to execute the reference method exactly as described when pursuing comparative evaluation studies. We demonstrated in this study that cross-reactivity does not occur with all master mixes. Three of the mixes that we evaluated showed no cross-reactivity with any non-target species. In addition, the BAM method primers and probe were evaluated for specificity on gDNA of additional species not available in our laboratory using both the Quantifast and Taqman Fast Advanced master mixes (*Cyclospora macaque*, *Cyclospora papionis*, *Entamoeba dispar*, *Entamoeba hystolitica*, *Entamoeba invadens*, *Neospora caninum* and *Plasmodium falciparum*), and no cross-reactivity was observed when the optimized protocol was used (da Silva, A., personal communication, Oct 2021). Therefore, a thorough evaluation of the master mix exclusivity is recommended for laboratories adopting the BAM method prior to implementation for diagnostic use, especially if alternative mixes are being considered.

In this study, the method performed reliably on produce samples that were up to 7 days old or frozen (either before or after spiking). This is consistent with a previous study that showed robust, reliable and sensitive performance of the BAM method on frozen berries [16]. We found that the proportion positive was lower for spring mix samples aged 7 d. Others have also reported an increase in Cq values and lower detection levels when using the BAM method on expired salad mixes [17]. Overall, the method performed favorably on aged samples, which may be significant in the case of a cyclosporiasis outbreak investigation, as epidemiological trace-back to linked food items can be a lengthy process, and often, only “leftover” samples may remain for testing. As freezing did not negatively impact detection in either the spring mix or raspberries, it may be preferable to preserve the implicated produce samples by freezing for later testing rather than refrigeration (which could lead to further deterioration of the sample matrix and/or contaminating oocysts); however, this requires more extensive validation.

Our repeatability experiments and proficiency panel results show that consistent performance can be achieved by multiple analysts. Regular proficiency testing is an ISO 17025 quality assurance requirement to ensure the generation of reliable results. Ideally, proficiency panels are prepared externally; however, due to the limited availability of *C. cayetanensis* oocysts and the lack of external providers, we have opted to prepare the panels in house. Poor access to oocysts may be a limiting factor for other laboratories wishing to verify and implement the BAM method for routine use; however, at minimum, use of the artificial positive control gene fragment could serve to fulfil this requirement.

The US-FDA Chapter 19b method does not require the sequencing of suspect positive amplicons as confirmation for reporting a *Cyclospora*-positive result. In previous diagnostic surveys, we performed sequencing confirmation for PCR-based methods. Therefore, we included sequencing in the diagnostic method for follow-up confirmation of a positive result. However, if sequencing fails, a sample may still be reported as positive if the Cq value is <42 or inconclusive if the Cq value is >42. It should be noted that, in order to sequence a suspect positive sample, the sample must undergo a second qPCR amplification without the IAC, as it would interfere with the sequencing. This second qPCR, if positive, also serves as a form of confirmation of the initial result, even if the sequencing subsequently fails.

Although sequencing of the qPCR amplicon would further support the confirmation of positive results, it is rarely successful for short fragments, such as those produced in this assay (<100 bp). During the verification of this method, we submitted multiple positive amplicons (generated from gDNA of stock *Cyclospora* oocysts or oocysts spiked on produce) for sequencing; many failed outright, and none resulted in the assembly of a contig longer than 50–60 bp. This partial fragment is from a conserved region of the 18S rRNA gene and, therefore, not unique to *C. cayetanensis*. The BLAST search results will therefore include many significant alignments to non-*Cyclospora* species. Thus, as a tool to confirm suspect-positive results, sequencing of the region targeted by this assay is unrewarding, as, even if a quality sequence fragment can be generated, it will not be very diagnostically informative. However, alignment, the alignment of any generated sequence with the artificial positive control (which contains known SNPs), can rule out cross-contamination. An ancillary assay targeting a longer fragment that is readily sequenced and genetically distinct could be employed for confirmation of the samples screened as positive by the BAM method. Temesgen et al. [12] developed a *C. cayetanensis* qPCR targeting a 141-bp product from the ITS-1 region that is amenable to sequencing and could be explored for this purpose.

The *Cyclospora* qPCR assay is a duplex assay utilizing primers and probes to target *C. cayetanensis* DNA and an internal amplification control (IAC) to enable the monitoring of each sample for the presence of PCR inhibitors that could prevent amplification of the target. The amplification of the IAC target is monitored in each sample well, and if the Cq value of the IAC in a test sample well is >3 Cq points above the average Cq of IAC for the NTC, the sample is considered to be inhibited and will be retested at a 1:4 dilution. We therefore performed verification testing of the method (see Table 2) on all matrix types anticipated to be tested for routine surveillance to ensure their performances were consistent in all targeted matrices, as different produce types may contain different inhibitory compounds. In all of the verification studies, only three samples showed inhibition in the IAC (Cq value > 3 points above the NTC). These were samples of raspberries for the robustness study spiked with 200 oocysts from the aged 7 days + frozen produce wash sub-set (Table 4). However, *Cyclospora* was still successfully amplified in those samples despite the indicated presence of inhibitors. This demonstrates the capacity of the test to detect *Cyclospora* despite inhibition; regardless, our method also prescribes follow-up testing of the diluted samples when inhibition is detected by the IAC to further optimize the detection of *Cyclospora.*

The number of *Cyclospora* positive sample detected rates in the first 15 months of implementing the BAM method for the CFIA’s food microbiological surveillance program far exceeded those for the previous decade of surveillance using the qPCR-MCA method. From 2011 to 2020, *Cyclospora* was detected in eight samples of produce out of approximately 11,569 samples tested during that period. Thus, 0.07% percent of the samples were positive, whereas using the BAM method enabled the detection of *Cyclospora* in 0.28% (5/1759) of samples. This difference can be primarily attributed to the increased diagnostic performance in the presence of molecular inhibitors of the BAM method compared to the previously employed qPCR-MCA method, although differences in the commodities tested may also be a factor. As discussed above, confirmatory sequencing of the short fragment generated for the positive samples by the BAM method is not generally successful or informative. There is therefore an urgent need for a confirmatory molecular epidemiological tool that performs reliably in *Cyclospora*-positive produce samples, which tend to contain relatively low concentrations of parasite DNA compared to human clinical samples. Such a tool could be used in surveillance or outbreak investigations as a follow-up confirmatory test for samples screened positive for *Cyclospora* by the BAM method and to further elucidate homologies between isolates recovered from clinical specimens and implicated foods. Cinar et al. [18] developed a workflow to obtain complete mitochondrial genome sequences from cilantro samples spiked with 200 *C. cayetanensis* for the typing of isolates; however, the utility of this method is limited if the sample contains multiple isolates of the parasite. A *C. cayetanensis* genotyping system based on eight genetic markers has been applied to human clinical samples [19]; however, this approach has not yet been employed successfully for food samples, and additional markers are required to improve cluster detection.

The results of the testing conducted under CFIA’s food microbiological surveillance program show that the vast majority of fresh imported produce available at retail in Canada is safe for consumption, but contamination by *Cyclospora* oocysts can occur in all seasons. Food-borne cyclosporiasis outbreaks in Canada have frequently been linked to berries and herbs [20], which yielded the highest proportion of samples positive in this study. In addition, three out of five produce samples positive for *Cyclospora* in this study were grown in the US, according to the package labelling. Although *Cyclospora* occurs worldwide, it has not generally been considered endemic in the US [21]. It is important to note that about two-thirds of the samples tested in the surveys were products of the US, which is more likely the reason for the higher number of *Cyclospora*-positive samples rather than a truly higher incidence compared to other countries. Others have adapted and implemented the BAM method for use in surveys of fresh produce. Barlaam et al. [22] also detected *C. cayetanensis* in berries collected in the Italian market using the BAM processing protocol and the 18S rDNA qPCR. The CFIA conducted appropriate follow-up actions for all of the positive samples identified in this study. No reported illnesses were linked to any of the affected products.

The molecular assays currently available cannot distinguish between live and dead organisms; therefore, the viability or infectivity of the oocysts detected on produce in this survey is not known. Regardless, the detection of *Cyclospora* DNA in produce should be considered a possible risk to public health for several reasons. First, the presence of *Cyclospora* DNA is an indicator of human faecal contamination somewhere in the food production chain, as there are no known animal hosts for the parasite. Second, while the sensitivity of the method is high, the rate of recovery of the oocysts from produce is not 100%, and populations of oocysts can consist of both viable and nonviable organisms. Therefore, even if there were reliable methods for determining that the recovered oocysts were nonviable and noninfective, the produce could still contain viable parasites and thus pose a potential risk to consumers.

## 5. Conclusions

In support of a harmonized approach to *Cyclospora* surveillance and disease outbreak investigations, we have verified and implemented the US-FDA BAM Chapter 19b method in our CFIA laboratory for routine diagnostic testing. Our verification studies have demonstrated that the method is sensitive, specific and performs robustly on a variety of matrices, including aged or frozen produce. Reliable and consistent results could be achieved by multiple analysts. CFIA’s food microbiological surveillance program results demonstrate the utility and fitness of the method and support its continued development as an ISO standard for the detection of *Cyclospora* in fresh produce.

## Figures and Tables

**Table 1 microorganisms-10-00559-t001:** Master mix exclusivity evaluation.

	qPCR Results (Cq Values for Replicate 1 and 2) for Master Mix				
Species	1	2	3	4	5
*Cyclospora cayetanensis*	+/+ ^1^ (34.24, 36.21)	+/+ (35.86, 39.45)	+/+ (33.68, 33.10)	+/+ (33.95, 35.85)	+/+ (35.85, 36.30)
*Eimeria acervulina*	−/− ^2^	+/+ (21.23, 20.33)	−/+ (N/A, 21.60)	−/−	−/−
*Eimeria maxima*	−/−	+/+ (22.83, 23.07)	+/+ (23.73, 23.84)	−/−	−/−
*Eimeria necatrix*	−/−	+/+ (23.38, 23.72)	+/+ (25.01, 25.57)	−/−	−/−
*Eimeria tenella*	−/−	+/+ (23.59, 22.49)	+/+ (25.5, 25.12)	−/−	−/−
*Eimeria bovis*	−/−	+/+ (23.04, 23.38)	+/+ (23.00, 22.79)	−/−	−/−
*Eimeria ahsata*	−/−	+/+ (19.72, 19.78)	+/+ (20.26, 20.94)	−/−	−/−
*Eimeria papillata*	−/−	−/−	−/−	−/−	−/−
*Isospora suis*	−/−	+/+ (25.02, 25.13)	+/+ (26.23, 26.47)	−/−	−/−
*Hammondia hammondi*	−/−	−/−	−/−	−/−	−/−
*Giardia duodenalis*	−/−	−/−	−/−	−/−	−/−
*Giardia muris*	−/−	−/−	−/−	−/−	−/−
*Toxoplasma gondii*	−/−	−/−	−/−	−/−	−/−
*Cryptosporidium parvum*	−/−	−/−	−/−	−/−	−/−
*Cryptosporidium hominis*	−/−	−/−	−/−	−/−	−/−
*Cryptosporidium meleagridis*	−/−	−/−	−/−	−/−	−/−

1. Quantifast (Qiagen). 2. Sso Advanced Universal Probes Supermix (Bio-Rad). 3. iTaq Universal Probes Supermix (Bio-Rad). 4. TaqMan™ Fast Advanced Master Mix (Applied Biosystems). 5. TaqMan™ Environmental Master Mix (Applied Biosystems). ^1^ Both replicates positive. ^2^ Both replicates negative.

**Table 2 microorganisms-10-00559-t002:** Number of positive samples detected by the *Cyclospora* qPCR for each matrix verified.

	Proportion Positive				DSe (%)			DSp (%)
Number of Oocysts Spiked	200	10	5	0	200	10	5	0
Matrix								
Raspberry	10/10	3/10	ND	0/4	100	30	ND	100
Strawberry	11/11	9/11	3/11	0/4	100	82	27	100
Cilantro	10/10	5/10	ND	0/4	100	50	ND	100
Romaine	10/10	2/10	ND	0/4	100	20	ND	100
Spring Mix	9/10	0/10	ND	0/4	90	0	ND	100
Kale Salad	9/10	5/10	ND	0/4	90	50	ND	100
Coleslaw	9/10	3/10	ND	0/4	90	30	ND	100
Blueberries	10/10	2/10	ND	0/4	100	20	ND	100
Green onion	9/10	3/10	ND	0/4	90	30	ND	100

ND = Not done.

**Table 3 microorganisms-10-00559-t003:** Impact of produce and wash storage time or condition on the proportion positive for the spring mix and raspberries.

	Produce Storage Time/Condition after Oocyst Spiking					
Produce Wash Storage Condition (Post-Processing)	Fresh		Frozen		7 days	
	Proportion Positive and Cq (Mean ± SEM)					
	Spring Mix	Raspberries	Spring Mix	Raspberries	Spring Mix	Raspberries
1 day (overnight at 4 °C)	5/5	5/5	5/5	5/5	4/5	5/5
	37.36 ± 0.80	35.36 ± 0.29	36.47 ± 0.60	35.83 ± 0.27	37.00 ± 0.79	37.46 ± 0.53
3 days (4 °C)	5/5	5/5	5/5	5/5	5/5	5/5
	36.68 ± 0.15	35.11 ± 0.51	36.24 ± 0.23	35.44 ± 0.43	37.41 ± 0.24	36.62 ± 0.31
Frozen (−20 °C)	5/5	5/5	5/5	5/5	4/5	5/5
	36.66 ± 0.63	36.63 ± 0.93	36.61 ± 0.16	36.23 ± 0.31	37.15 ± 0.59	36.63 ± 0.27

**Table 4 microorganisms-10-00559-t004:** Impact of freezing on the detection of *Cyclospora* in raspberries.

	Fresh	Frozen
Oocysts Spiked	Proportion Positive and Cq (Mean ± SEM)	
200	8/8	8/8
	36.45 ± 0.98	36.64 ± 0.54
10	5/8	3/8
	37.64 ± 0.35	37.91 ± 0.75

**Table 5 microorganisms-10-00559-t005:** Cq values generated for the *Cyclospora* artificial positive control on five different plates conducted by five different analysts.

	Cq Value of Cyclospora Standard 10^5^				
Analyst	Replicate 1	Replicate 2	Mean	Standard Deviation	% CV ^1^
1	20.74	21.01	20.88	0.19	0.91
2	20.63	21.14	20.89	0.36	1.73
3	20.37	21.17	20.77	0.57	2.72
4	21.37	21.11	21.24	0.18	0.87
5	20.80	21.13	20.97	0.23	1.11
		Overall	20.95	0.30	1.45

^1^ Coefficient of variation.

**Table 6 microorganisms-10-00559-t006:** Cq values for proficiency panels run by five different analysts.

	Analyst											
	1		2		3		4		5		Overall	
	Panel	Result (Cq)	Panel	Result (Cq)	Panel	Result (Cq)	Panel	Result (Cq)	Panel	Result (Cq)	Mean (SD)	%CV
Cilantro	+ ^1^	35.39	+	37.59	+	34.88	+	38.67	+	36.06	36.52 (1.58)	4.32
Strawberry	+	35.05	+	35.15	+	34.34	−	N/A	+	36.10	35.17 (0.72)	2.43
Raspberry	+	35.11	−	N/A	+	35.73	+	36.83	−	N/A	35.89 (0.87)	6.27
Spring Mix	+	35.05	+	35.59	−	N/A	+	37.86	+	35.30	36.45 (2.28)	3.43
Coleslaw	− ^2^	N/A ^3^	+	37.28	+	36.01	+	38.21	+	35.41	36.73 (1.26)	2.04
Romaine	+	36.75	+	35.91	+	35.50	+	36.65	+	35.04	35.97 (0.73)	3.67
										Overall	36.14 (1.33)	3.67

^1^ Spiked with 200 *C. cayetanensis* oocysts. ^2^ Un-spiked (negative). ^3^ Negative.

**Table 7 microorganisms-10-00559-t007:** Summary of the number and type of imported produce samples tested and confirmed positive for *C. cayetanensis*.

Sample Type	Number Tested	Number Positive for *Cyclospora* (% Positive)
Imported Leafy Greens and Salads (F263R)	352	1 (0.28%)
Imported Fresh Cut Leafy Vegetables and Vegetable Salads (SB3222)	181	0
Imported Fresh Whole Leafy Vegetables (SB3221)	120	0
Imported Fresh Leafy Herbs (SB3081)	489	2 (0.41%)
Imported Fresh Berries (SB341 and F227 ^1^)	597 (SB341) 20 (F227)	1 (0.17%) 1 (5.00%)
Total	1759	5 (0.28%)

^1^ Each submission for the F227 plan consisted of *n* = 5 samples from the same lot of berries, which were each tested individually.

**Table 8 microorganisms-10-00559-t008:** Summary of information on the samples positive for *Cyclospora*.

Sample Type (Sample Plan ID)	Month/Year Collected	Country of Origin	Average Cq Value 1st/2nd qPCR (Proportion of Replicates Positve)
Organic Mint (SB3081)	September 2020	Colombia	41.05/40.51 (3/3, 4/6)
Prewashed Baby spinach (F263R)	November 2020	United States	38.19/38.08 (3/3, 5/6)
Cilantro (SB3081)	February 2021	United States	44.06/44.00 (2/3, 1/6)
Blueberry (SB341)	June 2021	United States	39.60/39.58 (1/3, 2/6)
Raspberries (F227)	November 2021	Mexico	39.43/38.45 (2/3, 1/6)

## Data Availability

Not applicable.

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
