# Peer review of "Verification and Use of the US-FDA BAM 19b Method for Detection of Cyclospora cayetanensis in a Survey of Fresh Produce by CFIA Laboratory"

_microorganisms, 2022, doi:10.3390/microorganisms10030559_

Round 1

Reviewer 1 Report

Journal

Microorganisms (ISSN 2076-2607)

Manuscript ID

microorganisms-1596377

Type

Article

Title

Verification and use of US-FDA BAM 19b method for detection of Cyclospora cayetanensis in survey of fresh produce by CFIA laboratory

Authors

Laura Lalonde * , Jenna Oakley , Patrick Fries

Section

Parasitology

Special Issue

Cyclospora cayetanensis and Cyclosporiasis

Cyclospora cayetanensis, the causative agent of human cyclosporiasis, is an emerging protozoan pathogen that is endemic in developing countries and responsible for large foodborne outbreaks in North America since the 1990s, primarily associated with imported fresh produce from cyclosporiasis-endemic areas such as Guatemala and Mexico. To understand and mitigate human health risks posed by Cyclospora, produce growers and regulators require tools capable of assessing their presence. US-FDA’s BAM 19b was developed for the detection of Cyclospora, which includes sample preparation and a real-time PCR assay targeting the Cyclospora. BAM method had been validated by interlaboratory. In the current manuscript, the authors validated the BAM method to facilitate the harmonized surveillance and outbreak investigation of Cyclospora between the US and Canada. The authors examined the robustness of the BAM method in a variety of matrices including under adverse sample conditions such as 7 days aged samples, and freeze samples. However, a significant limitation of the BAM method using qPCR is its inability to differentiate between viable and nonviable parasites (as DNA can persist long after parasites die); this may inflate the perceived risk posed by such parasites in fresh produce (Table 4 and 5). 

Minor concerns:

  1. The scientific names of species need to be italicized throughout the text.
  2. Need more explanation why methods 2 and 3 in Table 1 are cross-reactive with closely related protozoa parasites, including Eimeria.

Author Response

Thank you very much for your helpful suggestions to improve the manuscript. Our responses to your comments are below:

  1. Lines 468-477 address the limitations of the assay with regards to distinguishing live vs dead parasites. 
  2. We have italicized all species names.
  3. Lines 342-363 provide some explanation on the cross-reactivity with some closely related parasites found with certain qPCR master mixes. 

Reviewer 2 Report

.

Author Response

Thank you very much for your helpful suggestions to improve our manuscript. Below are responses to your comments:

  1. The number of oocysts used in spiking experiments was modelled after the US-FDA interlaboratory validation study. Text has been modified and references included in lines 76 and 252 to clarify this point. Line 256 has also been edited to clarify the detection limit for spring mix.
  2. Species names have been italicized throughout.
  3. Lines 234-236 present the results of sequencing attempts and the authors consider this important to include in this section.
  4. The repeated information in Lines 262-266, 286-289 and 296-300 has been removed as suggested.
  5. Lines 387-396 provide explanation on the requirements for reporting a positive result. As indicated there, for samples with Cq values >42, sequencing confirmation was required for reporting as a positive (to rule out cross-contamination from the positive control), in addition to target amplification on test and re-test.  The number of positive replicates has also been included in Table 9. 
  6. Line 321 C. cayetanensis has been corrected.
  7. The references list has been updated to the MDPI journal style

Reviewer 3 Report

In this manuscript, authors adapted and verified the US-FDA’s BAM 19b method for detection of Cyclospora cayetanensis independently in their lab, and employed it in a national produce survey. The results are helpful for verifying the robustness of this method for routine diagnostic testing of Cyclospora cayetanensis.

A few suggestions are as follows.

The genus and species name of parasites should be italicized, e.g., C. cayetanensis, E. acervuline, …… in Lines 221-223, 234, 238, 254……

Table 2: ND, why so many samples spiked with 5 oocysts were not done?

Table 3 and Table 4 should be combined into one table.

Table 6 and Table 7: the title is too simple and not understandable by itself. Besides, legends should be included underneath the table, e.g., CV, +, -, N/A.

Discussion is too long. Some parts are redundant.

Author Response

Thank you for your helpful suggestions to improve our manuscript. Our response to your comments are below:

  1. Species names have been italicized throughout.
  2. The detection limit was reached for most produce types at the 10 spike level, so we did not perform spiking at the 5 oocyst level. 
  3. We have combined tables 3 and 4 as suggested and re-numbered all subsequent tables.
  4. The titles of table 5 and 6 have been updated and footnotes included.
  5. We have opted not to shorten the discussion as it was not noted by the other reviewers to be redundant or too long. A concise summary is provided in lines 474-481 as an alternative for some readers. 

Round 2

Reviewer 1 Report

Thank you for responding to all the concerns..